# Prospective Evaluation over 15 Years of Six Breast Cancer Risk Models

**DOI:** 10.3390/cancers13205194

**Published:** 2021-10-16

**Authors:** Sherly X. Li, Roger L. Milne, Tú Nguyen-Dumont, Dallas R. English, Graham G. Giles, Melissa C. Southey, Antonis C. Antoniou, Andrew Lee, Ingrid Winship, John L. Hopper, Mary Beth Terry, Robert J. MacInnis

**Affiliations:** 1Cancer Epidemiology Division, Cancer Council Victoria, Melbourne 3004, Australia; sherly.li@cancervic.org.au (S.X.L.); Roger.Milne@cancervic.org.au (R.L.M.); Dallas.English@cancervic.org.au (D.R.E.); graham.giles@cancervic.org.au (G.G.G.); Melissa.Southey@monash.edu (M.C.S.); 2Centre for Epidemiology and Biostatistics, University of Melbourne, Melbourne 3010, Australia; j.hopper@unimelb.edu.au; 3Medical Research Council Epidemiology Unit, University of Cambridge, Cambridge CB2 0QQ, UK; 4Precision Medicine, School of Clinical Sciences at Monash Health, Monash University, Melbourne 3800, Australia; tu.nguyen-dumont@monash.edu; 5Department of Clinical Pathology, University of Melbourne, Melbourne 3010, Australia; 6Centre for Cancer Genetic Epidemiology, Strangeways Research Laboratory, Department of Public Health and Primary Care, University of Cambridge, Cambridge CB1 8RN, UK; aca20@medschl.cam.ac.uk (A.C.A.); ajl65@medschl.cam.ac.uk (A.L.); 7Department of Genomic Medicine, Royal Melbourne Hospital, Melbourne 3050, Australia; Ingrid.Winship@mh.org.au; 8Department of Medicine, Royal Melbourne Hospital, University of Melbourne, Melbourne 3050, Australia; 9Department of Epidemiology, Mailman School of Public Health, Columbia University, New York, NY 10032, USA; mt146@cumc.columbia.edu

**Keywords:** breast cancer, risk model, discrimination, calibration

## Abstract

**Simple Summary:**

Australia has one of the world’s highest breast cancer incidences. For most women who are not classified as at high risk, eligibility and frequency of breast cancer screening in Australia is based solely on their age. Breast cancer risk models can help to optimise early detection and management of breast cancer. We evaluated six commonly used models over 15 years follow-up using an Australian community-based cohort of 7608 women aged 50–65 years. The BOADICEA and IBIS models best discriminated women who were at higher risk of developing breast cancer from those at lower risk, but no model apart from BOADICEA accurately predicted absolute risk across the risk spectrum. The BOADICEA model could be of clinical use for women of similar demography.

**Abstract:**

Prospective validation of risk models is needed to assess their clinical utility, particularly over the longer term. We evaluated the performance of six commonly used breast cancer risk models (IBIS, BOADICEA, BRCAPRO, BRCAPRO-BCRAT, BCRAT, and iCARE-lit). 15-year risk scores were estimated using lifestyle factors and family history measures from 7608 women in the Melbourne Collaborative Cohort Study who were aged 50–65 years and unaffected at commencement of follow-up two (conducted in 2003–2007), of whom 351 subsequently developed breast cancer. Risk discrimination was assessed using the C-statistic and calibration using the expected/observed number of incident cases across the spectrum of risk by age group (50–54, 55–59, 60–65 years) and family history of breast cancer. C-statistics were higher for BOADICEA (0.59, 95% confidence interval (CI) 0.56–0.62) and IBIS (0.57, 95% CI 0.54–0.61) than the other models (*p*-difference ≤ 0.04). No model except BOADICEA calibrated well across the spectrum of 15-year risk (*p*-value < 0.03). The performance of BOADICEA and IBIS was similar across age groups and for women with or without a family history. For middle-aged Australian women, BOADICEA and IBIS had the highest discriminatory accuracy of the six risk models, but apart from BOADICEA, no model was well-calibrated across the risk spectrum.

## 1. Introduction

Breast cancer is the most common cancer and cause of cancer death for women worldwide, with approximately 2.3 million incident cases in 2020, and carries a substantial burden of disease [1,2]. Australia and New Zealand have some of the highest age-standardised incidence rates globally (95.5 per 100,000), twice the world average (47.8 per 100,000) [1]. 

Breast cancer risk models are currently used in familial cancer clinics by genetic counsellors to stratify women and inform risk-tailored advice on the optimal age range, frequency, and modality of screening for those at high risk [3,4]. For most women who are not classified as high-risk, eligibility and frequency of breast cancer screening in Australia is based solely on age. Risk models that incorporate pertinent risk factors that are easily obtained from questionnaires, including a woman’s breast cancer family history and lifestyle factors, may provide valuable information for this group of women to optimise early detection and management. Previously, we demonstrated the potential for two models (International Breast Cancer Intervention Study model (IBIS) and the breast and ovarian analysis of disease incidence and carrier estimation algorithm model: BOADICEA) to estimate 10-year risks using an Australian prospective cohort study [5]. Other commonly applied and validated risk models exist (the breast cancer risk assessment tool (BCRAT) [6], individualised coherent absolute risk estimators—literature model (iCARE-lit) [7], and BayesMendel (BRCAPRO and BRCAPRO-BCRAT) [8]), but there is a lack of studies comparing their performance, particularly over longer follow-up durations (>10 years), which is important for individual and population health care planning for conditions that have a long lead time [9]. We therefore aimed to evaluate and compare 15-year risk estimates for developing breast cancer across these six risk models. 

## 2. Materials and Methods

### 2.1. Study Design and Participants

The Melbourne Collaborative Cohort Study (MCCS) is a prospective cohort that includes 24,469 women from Melbourne, Australia, aged between 27 and 76 years (99% were 40–69 years) at recruitment [10]. All participants were of White European descent, including 8% born in Italy, 6% in Greece, and 8% in the UK/Malta with the rest born in Australia or New Zealand in our final sample. They attended baseline (1990–1994) and up to two waves of active follow-up in 1995–1998 and 2003–2007. Our analyses included women who were aged 50–65 years (with 65 years being the maximum age for 15-year risk estimation by age 80 years) when they attended follow-up 2 (2003–2007; designated as the start of follow-up for this analysis), since follow-up 2 had the most complete data on family history available. Women were eligible if they had completed the baseline and follow-up 2 questionnaires and had no prevalent breast or ovarian cancer prior to their follow-up 2 visit. The final sample used to evaluate models for 15-year risk of breast cancer consisted of 7608 women, 351 of whom were diagnosed with a first invasive breast cancer within 15 years after their follow-up 2 visit. For comparison, we also evaluated the models for their 5-year and 10-year risks. MCCS participants provided informed consent, and the Cancer Council Victoria Human Research Ethics Committee approved the study [10]. 

### 2.2. Risk Assessment

We used the latest versions (at the time of analysis) of the risk models: BOADICEA version 5.0.0, using updated Australian incidence rates [11]; IBIS version 8b [12]; BCRAT version 4.1 [6]; iCARE-lit version 1 [7]; and BRCAPRO and BRCAPRO-BCRAT version 2.1–7 [8]. These models varied in their underlying age-specific incidences of breast cancer, and input variables (Appendix A). 

At follow-up 2, MCCS participants completed a questionnaire that asked about their demographic characteristics and lifestyle-related factors, including, for example, age, alcohol intake, age at menarche, parity, number of sisters, brothers, and children, age at first birth, menopausal status, and use of the oral contraceptive pill and menopausal hormone therapy. Summary family history data on affected relatives were obtained from questionnaires at follow-up 2 (first-degree relatives) and follow-up 1 (second-degree relatives). Data from the most recent questionnaires were used and supplemented with that from older questionnaires if unavailable. To reconstruct pedigrees, the following assumptions were made about the year of birth (YOB) of participants’ relatives: mothers and aunts (25 years before the participant’s YOB), grandmothers (50 years before the participant’s YOB), sisters (participant’s YOB), and daughters (25 years after the participant’s YOB). Missing ages for affected and unaffected mothers, aunts, and grandmothers were imputed to 70 years, whereas for sisters, they were imputed to the youngest of the participants’ age at follow-up 2 or aged 70 years (except for BRCAPRO and BRCAPRO-BCRAT, where the software imputed missing ages). Weight at follow-up 2 was measured to the nearest 100 g using a digital electronic scale, while height was measured at baseline to the nearest 1 mm, using a stadiometer. Body mass index (BMI) was defined as weight (kg) divided by height squared (m^2^). History of hyperplasia or benign breast disease were not available. As our aim was to compare models using solely family history and lifestyle risk factor information, results from germline genetic testing for pathogenic variants in *BRCA1* and *BRCA2* (or susceptibility genes) and mammographic density were not included in our analyses due to lack of availability of these data for most women. 

### 2.3. Outcome Assessment

Incident cases and vital status were ascertained from record linkage between the Victorian Cancer Registry, the Victorian Registry of Births, Deaths, and Marriages, the National Death Index, and the Australian Cancer Database. Cases were notified to the Victorian Cancer Registry with a first diagnosis of invasive breast cancer (3rd Revision of the International Classification of Diseases for Oncology code C50) during follow-up to 1 March 2019. 

### 2.4. Statistical Analysis

Follow-up began at follow-up 2 attendance and ended at: (i) diagnosis of invasive breast cancer, (ii) follow-up time reaching 15 years, (iii) age 80 years (maximum age for estimating risk in BOADICEA), or (iv) censor date of 1 March 2019, whichever came first. Expected cancer counts for the defined cohort were estimated for each model by summing the predicted risks over all eligible participants. Deaths from causes other than breast cancer were included as competing risks for all models except for two, BOADICEA and BRCAPRO-BCRAT, because they do not currently have an option to account for competing causes of death.

We compared the performance of the models with up to 15-year risk in terms of discrimination and calibration. Calibration was assessed by comparing the number of expected cases (E) within the cohort with the number observed (O). Model discrimination was assessed using a concordance statistic (C-statistic) [13] and plotting receiver operating characteristic (ROC) curves, accounting for incomplete follow-up, where 1 indicates perfect discrimination and 0.50 indicates discrimination no better than chance. 

The assessment of model calibration at the individual level was graphically represented from the model’s goodness of fit using the calibration belt routine. This method uses likelihood-based tests on a data-driven forward selection of polynomial regression models to assess the goodness of fit of the 15-year risk estimates from the six models, where a *p*-value < 0.05 indicates miscalibration [14]. Calibration by quintiles of 15-year risk were also plotted [15]. Model calibration and discrimination were also examined stratified by age (50–54, 55–59 and 60–65 years) and by whether the women had an affected first- or second-degree relative. We also examined model performance for 5-year and 10-year risk. Sensitivity analysis included additional censoring at diagnosis of ductal carcinoma in situ. Analyses were performed using Stata (version 16) and R (version 3.6.1). 

## 3. Results

The study sample consisted of 7608 Australian women with a mean age of 58.5 years and mean BMI of 27.2 kg/m^2^; 23% had a first- or second-degree family history of breast cancer (Table 1). The six risk models have different input variables (Appendix A) and predicted risk distribution (Appendix A). IBIS and BOADICEA had the widest range of predicted-risk distribution.

The overall discrimination of 15-year breast cancer risk (measured by C-statistic) across the six models ranged between 0.51 and 0.59 (Table 2; Figure 1). IBIS and BOADICEA had higher discriminatory accuracy than the other four models; C-statistics were 0.57 (95% confidence interval (CI): 0.54,0.61) and 0.59 (95%CI: 0.56,0.62), respectively (*p*-difference compared with the other four models ≤0.04). C-statistics did not vary significantly across different age subgroups, whereas C-statistics from all models (except for BCRAT) were slightly higher for women who had a family history of breast cancer than for those who did not have any affected relatives. 

Overall summary measures of calibration showed that BRCAPRO and BRCAPRO-BCRAT overestimated risk (both E/O = 1.11, 95% CI: 1.00,1.23) (Table 2), particularly for those aged 60–65 years. Across the full spectrum of predicted risks, all models except for BOADICEA showed evidence of miscalibration (*p*-value < 0.03), where they generally underpredicted risk at the low end of risk and overpredicted risk at high end of risk (Figure 2). 

Findings were similar for 5-year and 10-year risk of breast cancer, except that only BCRAT and iCARE-lit showed evidence of miscalibration with 5-year risk (*p*-value < 0.02), whilst iCARE-lit had higher discrimination at 5-year and 10-year risk compared with 15-year risk (Table 3; Appendix A). Sensitivity analysis that included censoring for in situ breast cancer (91 cases) gave similar discrimination and calibration results (Appendix A).

## 4. Discussion

Of the six risk models assessed over a 15-year follow-up period in an Australian cohort of 7608 women, IBIS and BOADICEA showed superior discrimination between cases and non-cases compared with the other four models. All models except for BOADICEA showed evidence of miscalibration across the risk spectrum. 

Models that include multigenerational family history (IBIS, BOADICEA) had the widest range of predicted risk distribution. The exceptions to this wider predicted risk distribution by pedigree-based models were the two BRCAPRO models, likely because they only allow for the effects of *BRCA1* and *BRCA2* pathogenic variants in modelling familial relative risk, thus substantially underestimating the contribution of family history to the disease risk [14]. Interestingly, the predicted-risk distribution of BRCAPRO-BCRAT remained closer to BRCAPRO than BCRAT, probably because it adopts the same approach as BRCAPRO to estimate risk from family history. On the other hand, the risk estimated from BCRAT is not dependent on the likelihood of being a *BRCA1* or *BRCA2* mutation carrier, and the relative risks attributed to family history of breast cancer remain constant with age of the consultee. 

A previous study that analysed a combined Australian and North American cohort reported higher 10-year C-statistics for BCRAT (0.64), BRCAPRO (0.62), IBIS (0.66), and BOADICEA (0.65) [9]. This may be due to the study sample having a higher underlying risk, given that they were recruited from breast cancer family registries and had a much wider age range. Interestingly, in our stratified analysis for 15-year risk, we noted slightly higher C-statistics for those with a family history of breast cancer (Table 2) for all models except BCRAT. Similar measures of discrimination (area under the receiver operating characteristic curves (AUCs)) were reported for 6-year breast cancer risk within an American screening cohort aged 50 years or older for IBIS (0.60), but they detected higher AUCs for BCRAT (0.61) and BRCAPRO (0.58) [15]. It is unlikely that the lower discrimination results observed for BCRAT and the BRCAPRO models in our study were due to using underlying USA incidence rates (there was no option to select Australian rates); we observed virtually no difference in results when selecting Australian or USA incidence rates for the BOADICEA model (data not shown). The American study noted a similar overprediction using BRCAPRO for those without a family history of breast cancer and underprediction for those with >2 first/second degree family members [15]. Our estimated discriminatory ability for iCARE-lit was similar to that from a study of 15 international average-risk cohorts (including the MCCS), which reported a 5-year risk area under the ROC curve of 0.57 [7]. That study reported an underestimation of breast cancer for MCCS participants in the highest decile of 5-year risk using iCARE-lit [7], whereas we detected an overestimation of breast cancer risk in the highest quintile for 5-year risk (Appendix A). The previous study, however, had used MCCS baseline data; thus, it was less contemporary, participants were younger on average, and detailed family history information was not collected at baseline. A systematic review of validation studies of models including IBIS and BCRAT using average risk women outside of Australia have shown comparable moderate discriminatory accuracy [16].

IBIS and BOADICEA consistently outperformed other risk models in discrimination, including in family registries that encompass participants from Australia, USA, and Canada [9]. We show that their discriminatory performance is consistent over short and long periods of follow-up (5, 10, and 15 years) (Table 3). Additionally, calibration across different categories of predicted risk appeared relatively stable for BOADICEA when comparing 5-year, 10-year, and 15-year risk estimates. On the other hand, IBIS showed evidence of miscalibration for 10-year and 15-year risk. Results presented here on the calibration and discriminatory ability of the models could be used to determine the use of such models for long-term health planning, but clinicians should bear in mind the purpose and audience when deciding the most appropriate timeframe to estimate risk [17]. Women of child-bearing age may find shorter periods (e.g., 5-year) helpful when considering risk mitigation behaviours or therapy (e.g., mastectomy) that may affect family planning, whereas longer-term risk estimations (e.g., 15-year) may be useful for women at higher risk seeking earlier prevention [17]. Thus far, validation of risk estimates beyond 10 years has been limited by a lack of studies with long-term follow-up, so our findings fill a gap to support expansion of the clinical utility of BOADICEA and further evaluation of other models. 

We have previously demonstrated a doubling of discriminatory accuracy for IBIS and BOADICEA (C-statistic from 0.56–0.57 to 0.62) with the addition of a polygenic risk score to predictors examined (age, family history, and lifestyle factors) using a subsample of the MCCS [5]. These results were also in line with published data from the UK [18]. Additionally, Nguyen and colleagues used an agnostic approach to predict breast cancer with mammographic imaging and showed similar improvements in discrimination (AUC 0.63) [19]. Although the study samples are not directly comparable, this supports the view that the performance of risk models will be enhanced by the inclusion of input variables such as common genetic susceptibility variants and mammographic imaging-based measures.

## 5. Conclusions

We evaluated six breast cancer risk models within an Australian average-risk cohort of women aged 50 to 65 years and found that IBIS and BOADICEA had the highest discriminatory accuracy and that their discriminatory accuracy remained consistent over time. However, apart from BOADICEA, no model was well-calibrated across the risk spectrum. Breast cancer risk models can help strengthen preventive efforts such as screening programs for average-risk Australian women via tailored surveillance advice. Models with lifestyle-related factors and family history will further benefit from the inclusion of information on genetics and mammographic density.

## Figures and Tables

**Figure 1 cancers-13-05194-f001:**
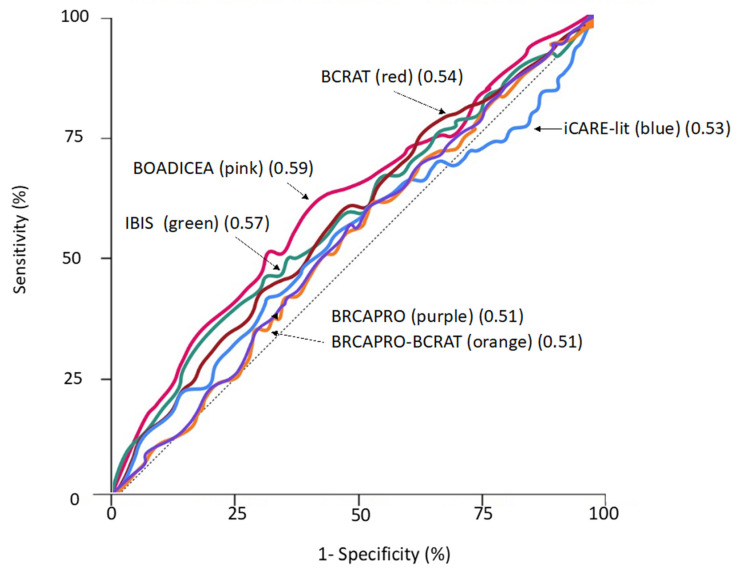
Receiver operating characteristic curves for six risk models. C-statistics are denoted in brackets. The dotted line denotes represents the line of no discrimination.

**Figure 2 cancers-13-05194-f002:**
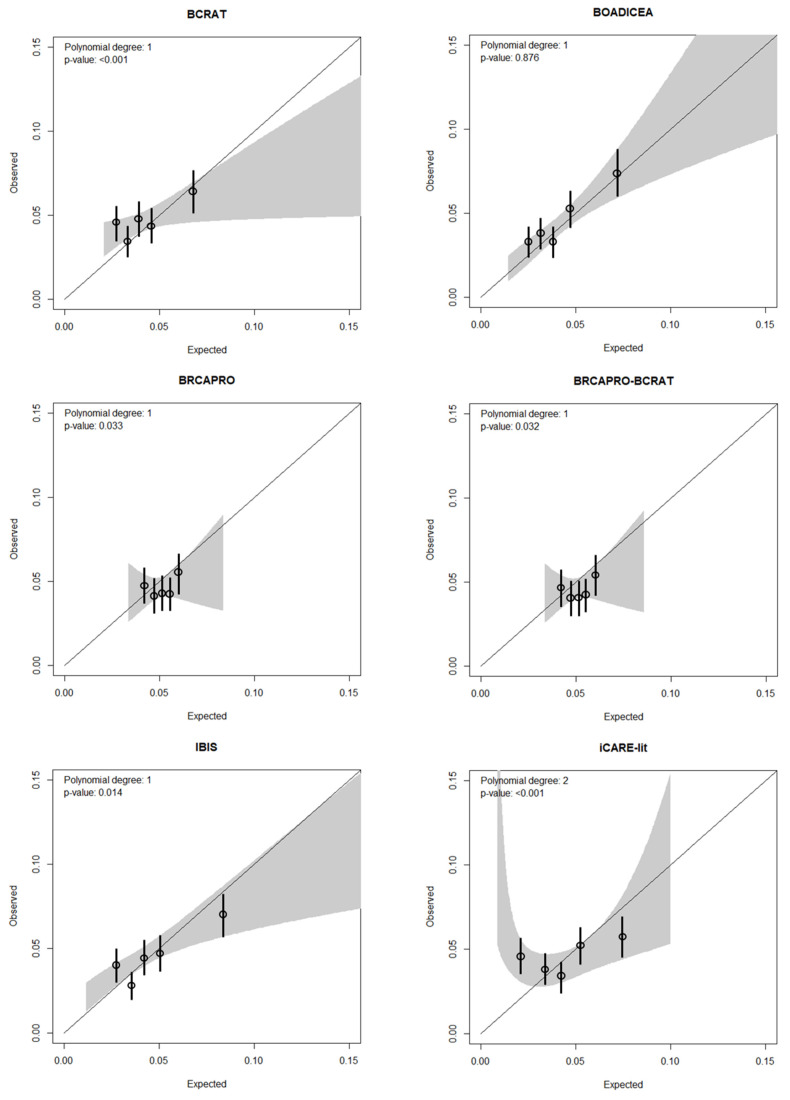
Calibration of 15-year breast cancer risk scores for six risk models across the risk spectrum.

**Table 1 cancers-13-05194-t001:** Characteristics of the Melbourne Collaborative Cohort Study participants (aged 50–65 years).

Characteristics	Mean	SD	
Age (years)	58.5	4.3	
Height (cm)	162.0	6.6	
Weight (kg)	71.3	13.9	
BMI (kg/m^2^)	27.2	5.3	
Alcohol intake (ethanol g/d)	8.9	11.9	
Menarche age (years)	12.9	1.6	
Number of live births	2.1	1.4	
Age at first birth (years)	25.4	4.8	
Age of menopause (years) ^1^	49.5	4.8	
Incidence of breast cancer per 1000 person-years ^2^	3.35 (95% CI: 3.01, 3.72)		
Characteristics	Number of women		**%**
Oral Contraceptive Use			
Never	1377		18.1
Former	6187		81.3
Current	37		0.5
Missing	7		0.1
Menopausal status ^3^			
Premenopausal	37		0.5
Postmenopausal	5962		78.4
Missing	1		0.0
Unable to determine	1608		21.1
Menopausal hormone therapy use ^4^			
Never	3848		50.6
Former	1643		21.6
Current Oestrogen	121		1.6
Current Oestrogen and Progesterone	752		9.9
Current hormone replacement therapy type missing	477		6.3
Missing ^5^	767		10.1
Family history of breast cancer ^6^			
No	5888		77.4
Yes	1720		22.6

Sample size: 351 cases, 7608 total participants; ^1^ Women whose reasons for periods stopping were due to having had a natural menopause or a bilateral oophorectomy; ^2^ Standardised incidence rate; ^3^ Postmenopausal is defined as: had menstrual period in last 12 months and currently using HRT (or missing) and aged at least 55 years; or no menstrual period in last 12 months (or missing) and periods stopped naturally; or no menstrual period in last 12 months (or missing) and periods stopped because ovaries were removed and two ovaries were removed; or no menstrual period in last 12 months (or missing) and periods stopped due to hysterectomy/other reason (or missing) and aged at least 55 years. ^4^ Type of hormone replacement therapy based on assumption of oestrogen for those who have had a hysterectomy and combined oestrogen and progesterone for those on HRT but have not had a hysterectomy. ^5^ Women in this category included those that were not asked or those where former use (between follow-up 1 and 2) could not be fully confirmed. ^6^ Family history in first- or second-degree relatives. SD: standard deviation; BMI: body mass index; cm: centimetres; kg: kilogram; g/d: grams/day.

**Table 2 cancers-13-05194-t002:** Calibration and discrimination of 15-year risks for six breast cancer risk models.

Risk Model	Number of Women	Expected Number of Cases	Observed Number of Cases	Expected/Observed Ratio (95%CI)	Concordance Statistic (95% CI)
Overall	7608				
IBIS		341.5	351	0.97 (0.88,1.08)	0.57 (0.54,0.61)
BOADICEA		342.4	351	0.98 (0.88,1.08)	0.59 (0.56,0.62)
BRCAPRO		389.3	351	1.11 (1.00,1.23)	0.51 (0.48,0.54)
BRCAPRO-BCRAT		389.7	351	1.11 (1.00,1.23)	0.51 (0.48,0.54)
BCRAT		327.9	351	0.93 (0.84,1.04)	0.54 (0.51,0.57)
iCARE-lit		339.5	351	0.97 (0.87,1.07)	0.53 (0.50,0.56)
Age 50–54 years	1912				
IBIS		90.0	91	0.99 (0.81,1.21)	0.59 (0.53,0.65)
BOADICEA		82.9	91	0.91 (0.74,1.12)	0.60 (0.54,0.66)
BRCAPRO		82.7	91	0.91 (0.74,1.12)	0.49 (0.43,0.55)
BRCAPRO-BCRAT		82.8	91	0.91 (0.74,1.12)	0.49 (0.43,0.55)
BCRAT		76.1	91	0.84 (0.68,1.03)	0.54 (0.48,0.60)
iCARE-lit		85.7	91	0.94 (0.77,1.16)	0.55 (0.49,0.62)
Age 55–59 years	2679				
IBIS		122.1	116	1.05 (0.88,1.26)	0.56 (0.50,0.61)
BOADICEA		124.2	116	1.07 (0.89,1.28)	0.59 (0.54,0.65)
BRCAPRO		134.9	116	1.16 (0.97,1.39)	0.54 (0.49,0.59)
BRCAPRO-BCRAT		135.0	116	1.16 (0.97,1.40)	0.54 (0.49,0.59)
BCRAT		114.7	116	0.99 (0.82,1.19)	0.58 (0.53,0.63)
iCARE-lit		120.4	116	1.04 (0.86,1.24)	0.51 (0.46,0.57)
Age 60–65 years	3017				
IBIS		129.4	144	0.90 (0.76,1.06)	0.58 (0.53,0.63)
BOADICEA		135.4	144	0.94 (0.80,1.11)	0.59 (0.54,0.64)
BRCAPRO		171.8	144	1.19 (1.01,1.40)	0.51 (0.46,0.56)
BRCAPRO-BCRAT		171.9	144	1.19 (1.01,1.41)	0.49 (0.44,0.54)
BCRAT		137.2	144	0.95 (0.81,1.12)	0.51 (0.46,0.56)
iCARE-lit		133.5	144	0.93 (0.79,1.09)	0.55 (0.50,0.60)
No family history of breast cancer ^1^	5888				
IBIS		217.7	241	0.90 (0.80,1.02)	0.54 (0.50,0.58)
BOADICEA		241.1	241	1.00 (0.88,1.13)	0.56 (0.52,0.59)
BRCAPRO		300.5	241	1.25 (1.10,1.41)	0.50 (0.46,0.54)
BRCAPRO-BCRAT		300.7	241	1.25 (1.10,1.42)	0.50 (0.46,0.54)
BCRAT		229.8	241	0.95 (0.84,1.08)	0.53 (0.49,0.56)
iCARE-lit		255.4	241	1.06 (0.93,1.20)	0.52 (0.48,0.56)
Family history of breast cancer ^1^	1720				
IBIS		123.8	110	1.13 (0.93,1.36)	0.57 (0.52,0.62)
BOADICEA		101.4	110	0.92 (0.76,1.11)	0.60 (0.55,0.65)
BRCAPRO		88.8	110	0.81 (0.67,0.97)	0.53 (0.47,0.58)
BRCAPRO-BCRAT		89.0	110	0.81 (0.67,0.97)	0.52 (0.47,0.58)
BCRAT		98.2	110	0.89 (0.74,1.08)	0.52 (0.46,0.57)
iCARE-lit		84.2	110	0.77 (0.63,0.92)	0.53 (0.47,0.59)

^1^ Family history in first- or second-degree relatives. IBIS: International Breast Cancer Intervention Study model (IBIS or Tyrer-Cuzick version 8b); BOADICEA: the Breast and Ovarian Analysis of Disease Incidence and Carrier Estimation Algorithm model (version 5.0.0); BRCAPRO: BayesMendel (version 2.1-7); BRCAPRO-BCRAT (version 2.1-7); BCRAT: the Breast Cancer Risk Assessment Tool (version 4.1); iCARE-lit: Individualised Coherent Absolute Risk Estimators—literature model (version 1); MCCS: Melbourne Collaborative Cohort Study; CI: confidence interval.

**Table 3 cancers-13-05194-t003:** Calibration and discrimination statistics for 5-year, 10-year, and 15-year risk.

Risk Model	Number of Women	Expected Number of Cases	Observed Number of Cases	Expected/Observed Ratio (95%CI)	Concordance Statistic (95% CI)
5-year risk	7608				
IBIS		121.8	124	0.98 (0.82,1.17)	0.57 (0.54,0.61)
BOADICEA		118.4	124	0.95 (0.80,1.14)	0.59 (0.56,0.62)
BRCAPRO		119.7	124	0.97 (0.81,1.15)	0.51 (0.48,0.54)
BRCAPRO-BCRAT		119.8	124	0.97 (0.81,1.15)	0.51 (0.48,0.54)
BCRAT		111.1	124	0.90 (0.75,1.07)	0.54 (0.51,0.57)
iCARE-lit		181.6	124	1.46 (1.23,1.75)	0.59 (0.56,0.62)
10-year risk	7608				
IBIS		245.7	252	0.97 (0.86,1.10)	0.58 (0.54,0.61)
BOADICEA		237.6	252	0.94 (0.83,1.07)	0.59 (0.56,0.62)
BRCAPRO		260.9	252	1.04 (0.92,1.17)	0.51 (0.48,0.54)
BRCAPRO-BCRAT		261.1	252	1.04 (0.92,1.17)	0.51 (0.48,0.54)
BCRAT		230.6	252	0.92 (0.81,1.04)	0.54 (0.51,0.57)
iCARE-lit		290.6	252	1.15 (1.02,1.30)	0.58 (0.55,0.61)
15-year risk	7608				
IBIS		341.5	351	0.97 (0.88,1.08)	0.57 (0.54,0.61)
BOADICEA		342.4	351	0.98 (0.88,1.08)	0.59 (0.56,0.62)
BRCAPRO		389.4	351	1.11 (1.00,1.23)	0.51 (0.48,0.54)
BRCAPRO-BCRAT		389.7	351	1.11 (1.00,1.23)	0.51 (0.48,0.54)
BCRAT		327.9	351	0.93 (0.84,1.04)	0.54 (0.51,0.57)
iCARE-lit		339.5	351	0.97 (0.87,1.07)	0.53 (0.50,0.56)

IBIS: International Breast Cancer Intervention Study model (IBIS or Tyrer-Cuzick version 8b); BOADICEA: the Breast and Ovarian Analysis of Disease Incidence and Carrier Estimation Algorithm model (version 5.0.0); BRCAPRO: BayesMendel (version 2.1-7); BRCAPRO-BCRAT (version 2.1-7); BCRAT: the Breast Cancer Risk Assessment Tool (version 4.1); iCARE-lit: Individualised Coherent Absolute Risk Estimators-literature model (version 1); MCCS: Melbourne Collaborative Cohort Study; CI: confidence interval.

## Data Availability

The MCCS data can be made available on request to pedigree@cancervic.org.au. The data are not publicly available because we do not have ethics approval to do so. Questionnaires are provided within the Appendix A. Databooks including summary of the data collected can be located here https://www.cancervic.org.au/research/epidemiology/health_2020, accessed on 30 July 2021.

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
