# Peer review of "Prospective Evaluation over 15 Years of Six Breast Cancer Risk Models"

_cancers, 2021, doi:10.3390/cancers13205194_

Round 1
Reviewer 1 Report
In the manuscript by Li et al. the authors organize prospective evaluation over 15 years of six breast cancer risk models. The article is comprehensive and provides an excellent analysis and overview of the currently available breast cancer risk models, which will provide valuable guidance and reference for clinicians and follow-up epidemiological research in the future. I would suggest only minor changes.
1. Although the author has made a detailed description of the contents of the questionnaire in the manuscript, however, I suggest that the original questionnaire could be presented in the manuscript as a figure or table, which will make readers more intuitive understand and learn the ideas of the authors in designing the questionnaire.
2. Page 1, line 30:"Australia has one of the world’s highest breast cancer incidence. " Citation or reference is missing.
3. Page 2, line 84-85:"including 12% born in Italy, 10% in Greece and 7% in the UK..." The data presented only accounts for 29%(12%+10%+7%) of the total, and the ancestral origin of the remaining 71% has not been clarified, and I don't see any significance in this data.
4. The manuscript needs linguistic improvement.
Reviewer 2 Report
This is a very important paper on a very important topic on the table since the 1980’s.
I have a few suggestions/questions:
- The strong point of the submitted paper is the 15 year follow up, consolidating the data at ten years of the cited CoAuthor MBTerry et al (Lancet Oncol 2019;20-504-17) on 18856 women from Australia, Canada and USA, comparing the same models with the same results about the better performance of BOADICEA and IBIS, suggesting an interesting idea: build an hybrid model incorporating the polygenic risk component of BOADICEA and the family-history factors included in IBIS model. This is particularly interesting also considering the very recent paper of Chi Gao et al (J Clin Oncol 39:2564-2573) assessing the joint association of pathogenic variants in breast cancer predisposition genes and polygenic risk scores with BC in the general population.
I suggest Authors cite and discuss the possibility of such an implementation - Wich are the models FDA licensed?
- Authors should also cite and discuss the systematic review of J Louro et al (British Journal of Cancer 2019;121:76-85) about the quality assessment of breast cancer risk models wich is not particularly positive
- So far it is impossible to put the test in any breast cancer screening. Wich is Author’s practical suggestion?
- Who is the Doctor deputed to perform the risk model (any) for a woman, explain results and suggest further clinical decisions?.
- Authors shoud also state and comment that BOADICEA, IBIS and BCRAT are available for free use on line. Is 'n't It dangerous?
Round 2
Reviewer 1 Report
Authors made correction according to my previous suggestions. Strongly recommend for publishing.